# The Development of a Novel Decision Support System for the Location of Green Infrastructure for Stormwater Management

**Jan K. Kazak** [1],* , **Jakub Chruściński** [2] **and Szymon Szewrański** [1]

[1]  Department of Spatial Economy, Wrocław University of Environmental and Life Sciences,
    ul. Grunwaldzka 55, 50-357 Wrocław, Poland; szymon.szewranski@upwr.edu.pl
[2]  GeoTechnologies Sp. z o. o., ul. Długosza 60, 51-162 Wrocław, Poland; kuba.chruscinski@gmail.com
*   Correspondence: jan.kazak@upwr.edu.pl or jankazak@gmail.com; Tel.: +48-071-320-5670

**Abstract:** In order to maximise the benefits of green infrastructure in a city's structure for urban adaptation to climate change, there is a need to support decision-makers in the urban design domain with adequate information that would help them to locate such green infrastructure in the most suitable places. Therefore, the aim of this study was to develop a novel decision support system (DSS) for the location of green infrastructure. The goal of the designed solution is to inform users about the location of urban hydrological sinks, which gather stormwater in urban watersheds, and the amount of water which could accumulate in each location depending on the defined precipitation and the soil's moisture conditions. The designed DSS is based on a multicomponent methodology including both atmospheric and soil conditions. The DSS was tested using a sample that presents the results of stormwater accumulation calculations. The obtained results show which green areas are the most suitable locations for green infrastructure solutions and which facility is optimal because of its retention abilities and amount of accumulated stormwater. The application of the designed DSS allows us to maximise the benefits of the implementation of green infrastructure within the existing urban land use. The fully editable component of hydrological conditions allows for testing projections of the potential amount of accumulated water in different precipitation scenarios. The study provides a DSS for use by local authorities which enables them to concentrate actions in order to better adapt cities to climate change and environmental extremes.

**Keywords:** decision support system; green infrastructure; stormwater management; sustainable urbanisation; urban adaptation; climate change; environmental management; what-if scenario analysis

## 1. Introduction

Climate change could have a significant impact on urban socio-environmental systems [1–3], which impact over half of the global population [4]. There are many different environmental aspects that influence the urban metabolism which are mainly related with temperature [5–8], atmospheric precipitation [9–13], or the synergy between climate elements and human activity [14,15]. In order to reduce the urban vulnerability to these phenomena, a new pathway for cities is promoted on the supranational level—one which is based on urban adaptation to climate change [16–18]. Urban adaptation to climate change refers to the ability to cope with more frequent and extreme weather events and everyday weather conditions [19]. In order to adapt our cities to climate change, the implementation of both technological and nature-based solutions should increase urban resilience to extreme weather events and the ability to deal with their unfavourable effects on the urban

metabolism [20,21]. Additional changes in the urban built environment can be made through different land use patterns [22–24]. The monitoring of changes in the built environment can be carried out by integrating new technologies [25,26] and can therefore help support decision-makers with detailed information presented in clear and visible ways [27]. The development of sustainable land use rules and guidelines for local authorities is an area of investigation that is currently receiving much attention in academia [28–33]. As a result, green investments should reduce environmental threats to human life and health as well as their properties [34–37], increasing the quality of life of citizens [38–40].

One of the solutions applied in urban adaptation to climate change is the implementation of green infrastructure within a city's structure. Green infrastructure is defined in the European Strategy on Green Infrastructure as *"a strategically planned network of high-quality natural and semi-natural areas with other environmental features, which is designed and managed to deliver a wide range of ecosystem services and protect biodiversity in both rural and urban settings"* [41]. This definition is relatively wide and encompasses many environmental aspects which can be classified as ecosystem services. However, among some groups of specialists, green infrastructure can be treated mostly as a solution for one major issue, while other functions are only supplementary. The United States Environmental Protection Agency stated that *"green infrastructure is a cost-effective, resilient approach to managing wet weather impacts that provides many community benefits"* [42]. It is an alternative to the single-purpose grey stormwater infrastructure, which is conventionally designed as a piped drainage and water treatment system to move urban stormwater away from the built environment. Green infrastructure reduces the amount of stormwater and treats it as a source while delivering environmental, social, and economic benefits [42].

## 2. Literature Review

The following section presents a review of the different benefits of green infrastructure solutions in urban structures as well as guidelines and tools which are developed in order to enhance the implementation of green infrastructure solutions.

As mentioned in the introduction, there are a number of benefits to green infrastructure which might be crucial in creating sustainable cities. The case of the United States Environmental Protection Agency shows that the main benefit is the mitigation of risks and losses caused by stormwater (moderation of extreme events in general), which is classified under regulation services. However, there are also other benefits to the implementation of green infrastructure. Additional regulating services may influence the local microclimate in terms of thermal comfort [43,44], carbon sequestration and storage [45,46], or waste water treatment [47]. Besides regulating services, there are also habitat or supporting services, like the restoration or preservation of ecosystems and their biodiversity [48,49] and the improvement of landscapes [50], as well as cultural services [51,52]. The synergy of some of those benefits leads to more complex benefits like public health [53], which go beyond the initial assumptions of green infrastructure investments. Different types of benefits may be noticed in specific locations based on local factors and types of green infrastructure investments [54]. What is more, it is proven that green infrastructure applications can effectively support city management processes even in less obvious issues like, for instance, reducing seismic vulnerability and energy demand [55]. Regardless of the benefits of green infrastructure in each case, all these applications lead to ecologising our cities [56].

However, depending on these local conditions, different benefits might play the most significant role in built environment regulation [57]. Even within the same city, spatial distribution and differences in green infrastructure forms can predesign some areas as being more suitable for recreational facilities, while other areas can be more beneficial due to potable water savings or greenhouse gas issues. A multicriteria assessment of such functions was carried out for eight sites in Melbourne (Australia) in order to evaluate their stormwater harvesting functions [58]. In the case of stormwater gathering, the crucial aspect would be the elevation of the city's surface. Based on urban watersheds, different locations would present different suitability for water retention. Green infrastructure would not replace classical stormwater systems; however, by mitigating some parts of the stormwater flow, there is a

chance to use the bio-retention abilities of green areas [59]. Therefore, due to the unique local conditions, there is no regular spatial pattern of the location of green infrastructure. This decentralised stormwater system has different clustering densities throughout a city's structure [60]. That is why decisions about the optimal locations of green infrastructure are more complicated and should not be made based only on two-dimensional information. In order to optimise green infrastructure governance, public authorities can gather feedback from citizens. In the case of decentralised and scattered elements like green infrastructure, citizens can be treated as local experts, as their experiences concerning local conditions are wider and more holistic when compared with centrally managed approaches. Such green infrastructure governance supporting public participation can be obtained by the use of e-tools [61].

Green infrastructure has become particularly widespread in recent years, and nowadays it is included in many urban redevelopment projects and city development strategies [62]. It is also more common that governments or local authorities actively stimulate the process of green infrastructure investments [63]. One such action in this direction is the development of guidebooks and instructions promoting green infrastructure solutions [64]. The advantage of such local guidebooks is the fact that they may include local conditions, like type of soil or species of plants that are more suitable for the local climate [65]. In the case of Philadelphia (USA), the guidebook presents the whole design flow, starting from available resources, through site evaluation and stormwater management practice evaluation, and finishing on the selection of plants. The guidebook contains a template for the site assessment checklist, a plant selection catalogue connected to the local climate and shading conditions, and a review of case studies and examples of projects [65]. Moreover, in the case of Wrocław (Poland), stormwater management practices are assessed according to their retention abilities, replacement costs, nuisance of operation, and water purification abilities (Figure 1). Such a unified framework makes the comparison of analysed solutions easier. Moreover, each solution contains templates of architectural projects (Figure 2), which are freely available, to facilitate urban designers and architects integrating it into their work.

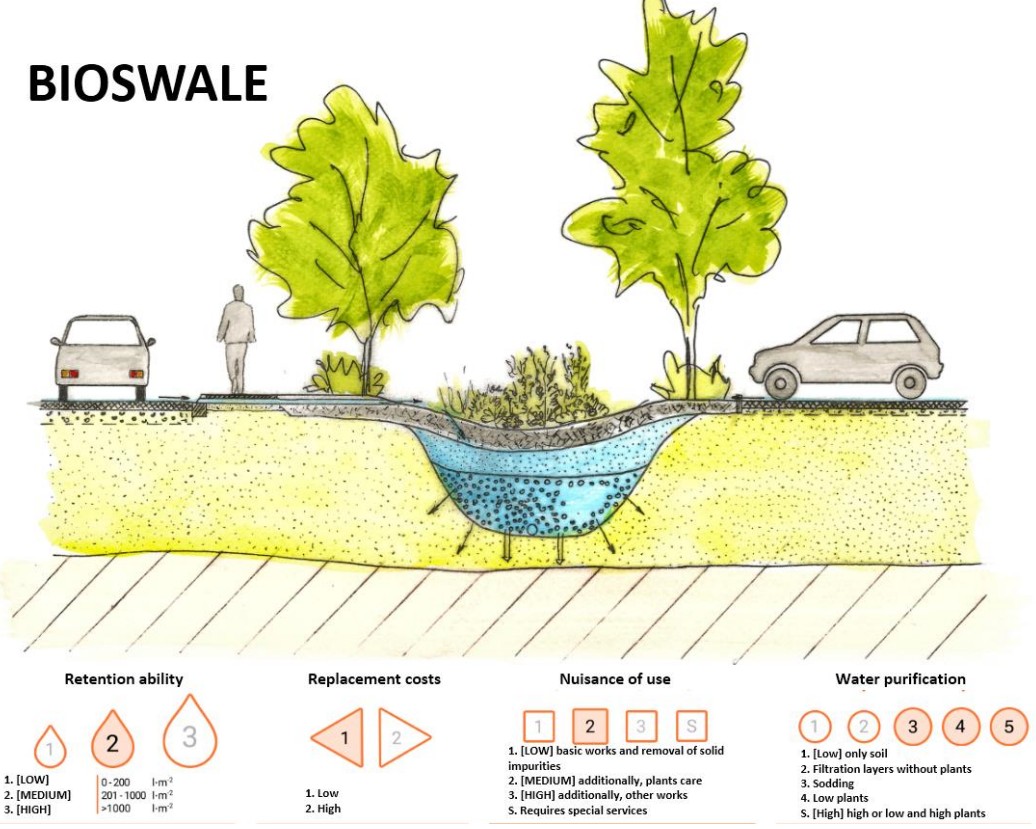

**Figure 1.** A case of stormwater management practice with a brief assessment [66].

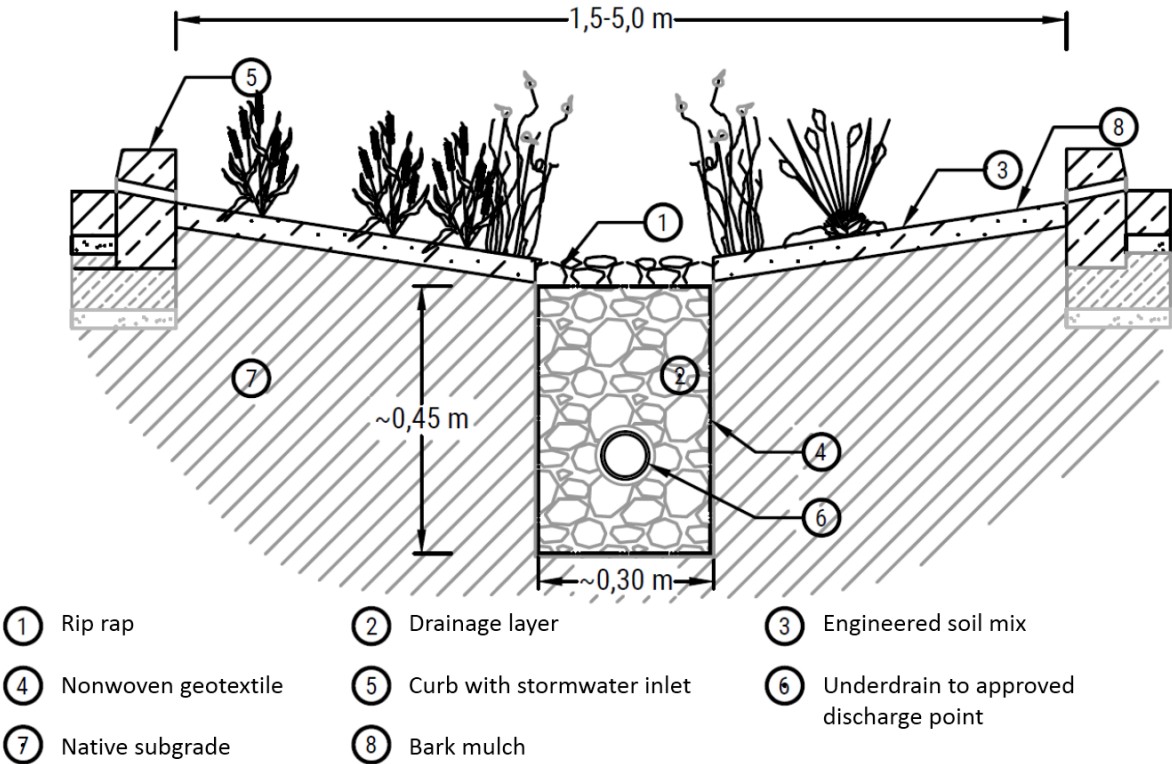

**Figure 2.** Template of architectural projects with ranges of possible sizes [66].

As presented in Figure 1, every solution has its individual retention abilities. That is why, besides the total elevation of the urban surface that was mentioned before, there is a need to analyse how much water can be accumulated in each location. This information is crucial in order to make smart decisions in city management while deciding about redevelopment projects and suitable locations for green infrastructure investments.

Therefore, in order to avoid unprofitable investments, there is a need to support decision-makers with information about the proper locations for specific green infrastructure investments. As Koutsoyiannis et al. [67] concluded in their work on a decision support system for the management of water resource systems, *"a well-known problem in development and application of a DSS is the general lack of communication between analysts/developers and decision-makers (or their technical support staff)"*. They also highlighted that very often too much attention is given to data processing techniques. Therefore, there is a need to link technical solutions with stakeholders to avoid data redundancy and to search for a solution that gives the knowledge to the decision-makers [68]. Most of the DSSs for water management operate on a scale that allows us to describe water management processes in a wider context. Therefore, they enable us to identify hot-spots of stormwater issues in a district of the city [67,69]. However, they fail to analyse local conditions, and the obtained results do not describe the situation in detail with the high precision of the input data. Therefore, in practice, we still face the problem of a lack of local solutions which would refer not to holistic approaches for bigger areas, but to everyday decisions that have to be made by public service units.

The in-depth literature review has proved the gap in the existing studies in the field of analysing optimal green infrastructure locations by taking into account stormwater management. The scope of this research is to integrate the best practices from the domains of hydrological and urban analyses. In order to support practice, the ultimate goal of this research is to develop solutions for the identification of some basic elements which might be helpful for effective decision-making processes in urban management. Besides the theoretical framework, the verification of the DSS on the sample will present the potential to answer, on the one hand, the question of how much water can flow

into an analysed location, and on the other hand, where the water accumulates due to the current surface elevation.

The aim of this research was to develop a novel decision support system (DSS) for green infrastructure location which can inform the user about the locations of urban hydrological sinks and the amount of water which could gather in each location depending on the defined precipitation conditions.

## 3. Materials and Methods

The designed DSS is based on a multicomponent methodology including atmospheric and soil conditions. The proposed solution is based on Geographical Information Systems (GIS), which are commonly used in hydrology in the context of surface run-off detection [70–72]. Hydrological modelling was based on the Soil Conservation Service Curve Number method (SCS-CN) [73,74]. The workflow of the system is presented in Figure 3 and explained in detail in the following subsections.

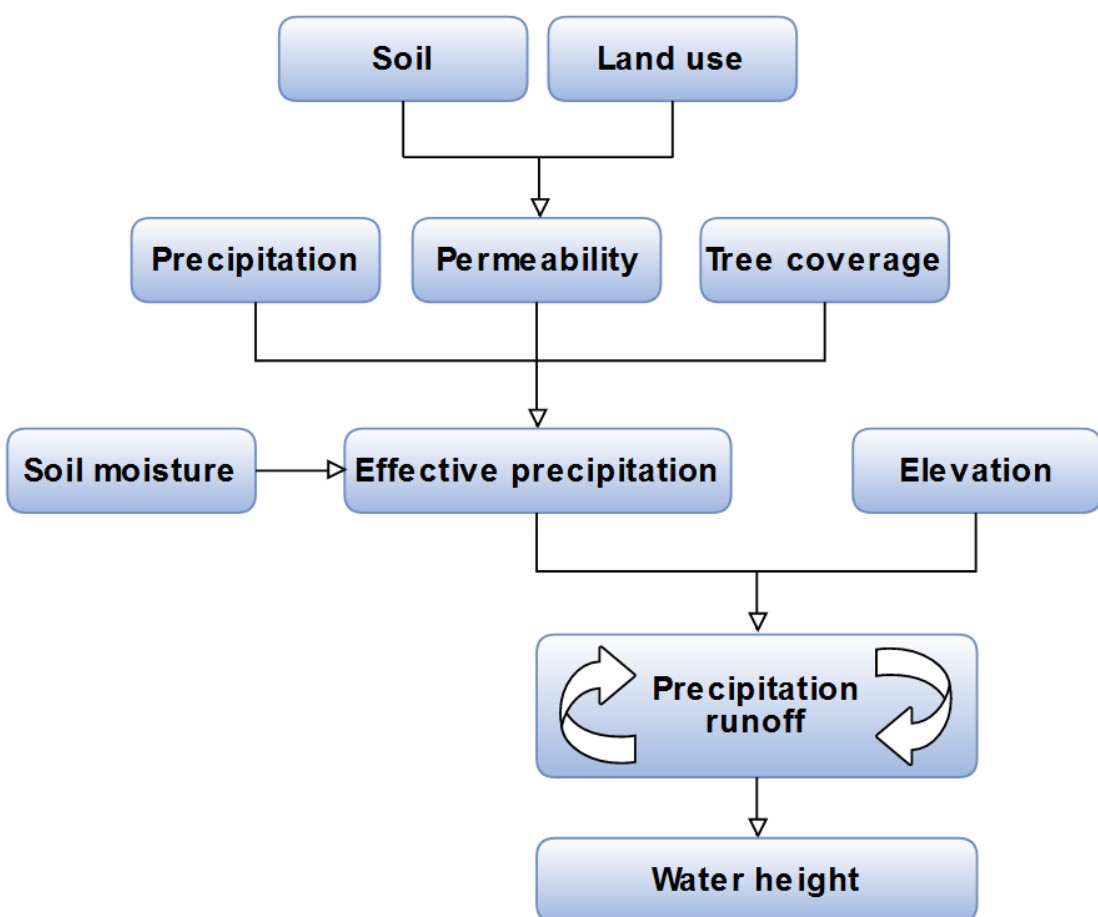

**Figure 3.** The workflow of the designed system.

The application of the designed DSS was conducted through a scenario-based approach in order to test a wide spectrum of options, including land use or land use with tree cover, different soil moisture conditions, and different modifications of precipitation (Table 1). All scenarios tested in this study are based on the "what-if" approach, which means that the results do not present the most common situations, but they verify the potential results of predefined settings. The scope of this study is to integrate the best practices from the hydrological and urban domains. The development of the novel DSS constitutes an innovation in the urban design domain by way of the applicable solution which unifies software which integrates a verified hydrological method (SCS-CN), well recognised among

specialists in the water management field, with GIS-based analyses which are commonly used in the domain of urban studies.

**Table 1.** Scenarios of tested land use and atmospheric conditions.

| Scenario | Precipitation 20 mm | Precipitation 60 mm | Precipitation 100 mm |
|---|---|---|---|
| Land use (soil moisture 2) | $S_1$ | $S_2$ | $S_3$ |
| Land use and tree cover (soil moisture 1) | $S_4$ | $S_5$ | $S_6$ |
| Land use and tree cover (soil moisture 2) | $S_7$ | $S_8$ | $S_9$ |
| Land use and tree cover (soil moisture 3) | $S_{10}$ | $S_{11}$ | $S_{12}$ |

All elements which vary in the analysed scenarios are elaborated in the following subsections. More detailed descriptions of the SCS-CN method are elaborated in previous articles [70,72]. In order to set realistic values and to verify the system with the use of real data, all calculations were carried out based on a case of a multifamily housing neighbourhood in Wrocław (Poland). Nevertheless, the assessment procedure is applicable for any given study area.

### 3.1. Verification Sample and Data

In order to verify the DSS, the test area was established. The verification sample allowed us to test the calculation process on real data and check the application potential. However, it is important to highlight that the same calculations can be performed for every area for which the data mentioned below are available.

The verification sample was a part of a multifamily housing area (Bartoszowice unit) in a Polish regional city (Wrocław). The sample represents a neighbourhood developed under two different urban design patterns. The southern part of the study area was developed at the beginning of the 1970s, while the northern part was developed at the end of the 1990s.

In order to support the designed DSS to calculate each case, there is a need to supply the following data:

1. Soil types—verification sample: Geological and engineering atlas of the Wrocław agglomeration [75]. This geological and engineering atlas presenting soil types was prepared based on thematic maps at the scale 1:10,000.
2. Land use—verification sample: The real estate cadastre register kept by the Municipal Board of Geodesy, Cartography and Cadastre. The accuracy of coordinates in that database is within 0.10 m.
3. Tree coverage and elevation—verification sample: Airborne laser scanning Light Detection and Ranging (LiDAR) for urban areas. Using LiDAR, it is possible to achieve a 0.5 m × 0.5 m resolution; however, in this research, a 1.0 m × 1.0 m grid was constructed.

All data were used in a Python script developed for ArcGIS. All input data used in the program were converted to a vector format, which is the basic data format used by the program. Mutual spatial relations among data were used to integrate the values required in the model from different input vectors.

At the stage of preprocessing the input data from the LiDAR point cloud data, measurement points corresponding to the soil class and buildings were selected. Next, based on the average values of points, a grid with dimensions of 1 m × 1 m was made to the raster with the same pixel size. The resulting Digital Elevation Model (DEM) was then used as an input to the iterative process of surface runoff analysis.

### 3.2. Soil

The first factor influencing the effective precipitation is the soil type. As the soil type influences the spatial variation of the distribution of volumetric moisture in soil, it is a basic element included in hydrological research [76]. The information about soil type was based on results of geological drilling. The study area is characterised by six different soil types, which has an impact on the retention abilities of the soil profile. According to the SCS-CN method, as soil class A, the following soils were classified: (1) anthropogenic soils, nonstructural embankments; (2) Pleistocene river noncohesive soils; (3) Holocene river noncohesive soils; and (4) Holocene organic unseparated lands. Two soil types in the study area were classified as B: (1) Holocene river organic soils and cohesive silts, and (2) Holocene river cohesive soils [77].

### 3.3. Land Use

The next element of stormwater accumulation is the land use. Depending on the surface cover, the relation between soil profile retention, surface accumulation, and evaporation is different. The database for the land use type was the local real estate cadastre system, which guarantees the most precise information that is needed in the case of spatial analyses on that scale. Therefore, the real estate cadastre supported our research with detailed locations of different land uses, with the precision of coordinates at the level of 0.10 m for building location points and boundaries of real estate plots [78]. In the analysed case study, the land use layer contained three classes: buildings, impervious roads and pavements, and green areas.

### 3.4. Permeability

The hydrologic soil group selected in Section 3.2 and land uses from Section 3.3 allowed for the definition of initial curve numbers (CN) (Table 2).

**Table 2.** Initial curve number (CN) selection.

| Land Use | Percent of Impermeability | Hydrologic Soil Group A | Hydrologic Soil Group B |
|---|---|---|---|
| Buildings | 100% | 89 | 92 |
| Impervious roads and pavements | 95% | 89 | 92 |
| Green areas | 10% | 32 | 58 |

For the studied area, four initial CNs describe the analysed site: 32, 58, 89, and 92. In order to include the percentage of impermeability, initial CNs were corrected by Equation (1) and presented by CN composites [77]:

$$CN_C = CN_I + \left( \frac{P_{imp}}{100} \right) (98 - CN_I), \tag{1}$$

where

$CN_C$—composite runoff curve number,

$CN_I$—initial runoff curve number,

$P_{imp}$—percentage of impermeability [%].

### 3.5. Precipitation

The next basic driver influencing stormwater accumulation included in the model is precipitation. This value can be freely defined by a user and may vary depending on local conditions. According to the Polish criteria for warnings of intense rainfall, characteristic thresholds are rainfall of over 30 mm, 50 mm, or 90 mm of precipitation [79]. Therefore, for the second and third benchmarks, precipitation at the levels of 60 mm and 100 mm was tested. However, based on local conditions and

the frequency of rainfalls with specific amounts of precipitation [80], the first variable was reduced to 20 mm of precipitation.

### 3.6. Tree Coverage

In order to increase the precision of modelling the surface type's influence on effective precipitation, 9 out of 12 scenarios also included tree coverage and its impact on water retention in tree crowns. In case of scenarios $S_1$–$S_3$, the use of land use without tree cover allowed for the preparation of reference scenarios. However, scenarios $S_4$–$S_{12}$ additionally include tree cover of the analysed area, which influences the precipitation conditions due to the amount of precipitation absorbed by tree crowns [81] and the trunk bark structure [82]. Based on a literature review, the amount of *throughfall* and *stemflow* can reach between 50% and 90% depending on species or intensity of precipitation [81,83,84]. Therefore, in the model, the amount of throughfall and stemflow was assumed at a level of 70%. That gives at least 30% stormwater absorption by tree crowns. For the purpose of the presented assessment, it was assumed that absorption is at the level of 30%.

Tree crowns were identified by LiDAR data and allowed for the preparation of more realistic model of the analysed case. The difference between land use in scenarios $S_1$–$S_3$ and land use with tree cover in scenarios $S_4$–$S_{12}$ is presented in Figure 4.

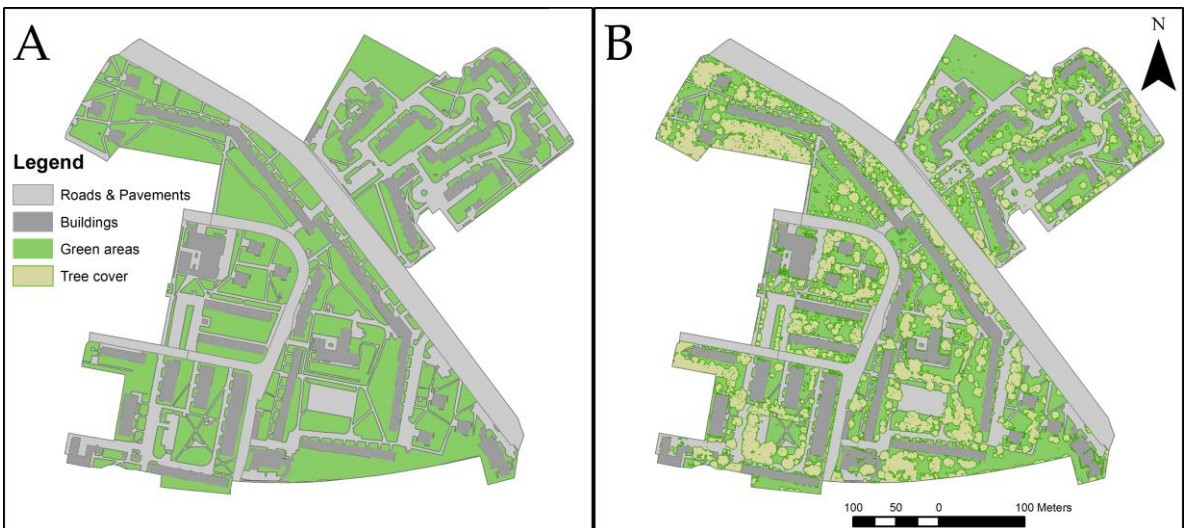

**Figure 4.** (**A**) Land use in the study area; (**B**) land use with tree cover in the study area.

### 3.7. Soil Moisture

An additional driver influencing scenario comparison is the soil moisture level, which is represented by the Antecedent Moisture Condition (AMC). The CN calculated above was adjusted by the AMC for three different moisture situations. Scenarios $S_1$–$S_3$ and $S_7$–$S_9$ were calculated based on AMC II, which represents regular soil moisture conditions. Scenarios $S_4$–$S_6$ represent dry conditions, and AMC I factors with values smaller than 1 reduce the values of CNs for each soil type. Finally, scenarios $S_{10}$–$S_{12}$ analyse wet conditions and include AMC III factors higher than 1. AMC factors allowed for the conversion of CNs from regular moisture conditions into other CNs which present different infiltration features. The AMC is a variable which is calculated in the DSS. Based on the field studies which gave the basis for the SCS-CN method, three different AMCs are defined in the standardised tables. Therefore, the AMC influenced the selection of variables which were taken from the standardised table [77].

### 3.8. Effective Precipitation

The calculation of the effective precipitation was based on the SCS-CN method, which was proposed by the United States Department of Agriculture in 1989. SCS-CN is an empirical model and was based on statistical analyses obtained from processing large experimental datasets. SCS-CN characterises the natural processes of stormwater runoff by considering the surface and its hydrologic conditions [69]. The SCS-CN model is one of the most popular runoff models and has been constantly developed and improved in many studies [73,85–87]. The basic formula in SCS-CN for calculating the runoff depth (Q) is expressed by Equation (2):

$$Q = \frac{(P - I)^2}{P - I + S} \qquad (2)$$

where

Q—direct runoff [mm],
P—total precipitation [mm],
I—initial abstraction [mm],
S—potential maximum retention [mm].
The potential maximum retention can be expressed by Equation (3):

$$S = 25.4 \left( \frac{1000}{CN} - 10 \right) \qquad (3)$$

where

25.4—conversion rate from English units to International System of units (from inches to milimetres),
CN—curve number.
The curve number is related to the soil characteristics, land cover, and initial soil moisture [72].

### 3.9. Elevation

In following steps, the effective precipitation calculated above represents the amount of stormwater accumulated in the study area. In order to identify that location, the elevation of the land surface was included. For that purpose, LiDAR data were used, and based on these data a DEM was created. Next, the DEM was divided into micro urban water catchments, which created hydrological sinks.

### 3.10. Precipitation Runoff

The precipitation runoff was modelled in an automatic iterative manner. As mentioned in Section 3.1, at the stage of preprocessing the input data from the LiDAR point cloud data, measurement points corresponding to the soil class and buildings were selected. Only such classes were able to constitute a physical barrier to water surface runoff. Then, based on the average value of points, a grid with dimensions of 1 m × 1 m was made to the raster with the same pixel size. This DEM was used as an input to the iterative process of surface runoff analysis. The effective precipitation flowing on the DEM surface depends on the slope direction and on filling out each hydrological sink. The DEM input grid performs multiple tasks: Firstly, the vector layer generated on it from the centroids of individual pixels provides information about the altitude value for the first interception. Once a hydrological sink is full (the water level reaches the edges of each sink), the system creates a new DEM$_i$ which excludes that hydrological sink from further iteration steps. At this stage, that hydrological sink is covered with a plane surface and stormwater is allowed to flow further to the next sinks according to the DEM flow direction. These steps are repeated until the effective precipitation has been used up. Finally, precipitation runoff DEMs constitute the output for the calculation of stormwater volume.

*3.11. Water Height*

In order to calculate the accumulated stormwater in each location that can be managed by green infrastructure, the initial DEM was deducted from the final DEM which gave the water height in each part of the study area (Equation (4)).

$$H_{rainwater} = \left( DEM_{final} - DEM_{initial} \right) [mm] \tag{4}$$

## 4. Results

*4.1. Stormwater Accumulation*

Based on the designed DSS for the location of green infrastructure, the results of the calculations present areas for the case study where stormwater gathering occurs in every analysed scenario. The results are presented in Figure 5. The output of the analysis is the raster layer, which can be displayed on the land use information. The numerical results of the analyses are presented in Table 3.

**Table 3.** Results for stormwater accumulation scenarios.

| Scenario | Area in Water Level Classes [m$^2$] | | | | Volume [dm$^3$] | Max Water Height |
|---|---|---|---|---|---|---|
| | 1.1–6.0 | 6.1–17.0 | 17.1–60.0 | 60.1–99.0 | | |
| $S_1$ | 15,130 | 2394 | 76 | 0 | 64,869 | 27 |
| $S_2$ | 31,013 | 16,323 | 2334 | 3 | 321,670 | 62 |
| $S_3$ | 38,668 | 28,348 | 8285 | 45 | 631,405 | 99 |
| $S_4$ | 15,386 | 2328 | 77 | 0 | 64,088 | 26 |
| $S_5$ | 24,816 | 13,854 | 1852 | 3 | 264,625 | 62 |
| $S_6$ | 31,074 | 22,794 | 6419 | 39 | 500,956 | 95 |
| $S_7$ | 14,355 | 2204 | 59 | 0 | 60,620 | 26 |
| $S_8$ | 28,668 | 14,912 | 2025 | 3 | 291,050 | 62 |
| $S_9$ | 36,421 | 25,681 | 7307 | 41 | 569,334 | 96 |
| $S_{10}$ | 14,802 | 2246 | 59 | 0 | 61,825 | 27 |
| $S_{11}$ | 33,385 | 16,802 | 2320 | 4 | 333,170 | 63 |
| $S_{12}$ | 40,291 | 30,129 | 8583 | 45 | 663,834 | 99 |

The results showed that with precipitation at the level of 20 mm, it is possible to select areas of the total case study area with stormwater accumulation higher than 60 mm. That result shows the potential for green infrastructure locations in this area. Two locations in the north-eastern part of the case study, zoomed in Figure 5, are the locations of stormwater gathering. These locations are at the edge of the road or pavement and the green areas. Their sizes are (1) 422 m$^2$ and (2) 589 m$^2$, with the highest water levels at (1) 24 mm and (2) 21 mm. In order to increase the efficiency of green infrastructure development within the city structure, these two locations should be taken into account. Additionally, it is possible to measure the whole volumes of stormwater in these locations, which are at the levels of (1) 2640 dm$^3$ and (2) 3114 dm$^3$. Therefore, the designed model is not only helpful for choosing a suitable location, but it also supports decisions about exact green infrastructure solutions that suit the local area based on possible stormwater resources. This feature of the model corresponds with the information that is included in green infrastructure good practice catalogues, like the example presented in Figure 1.

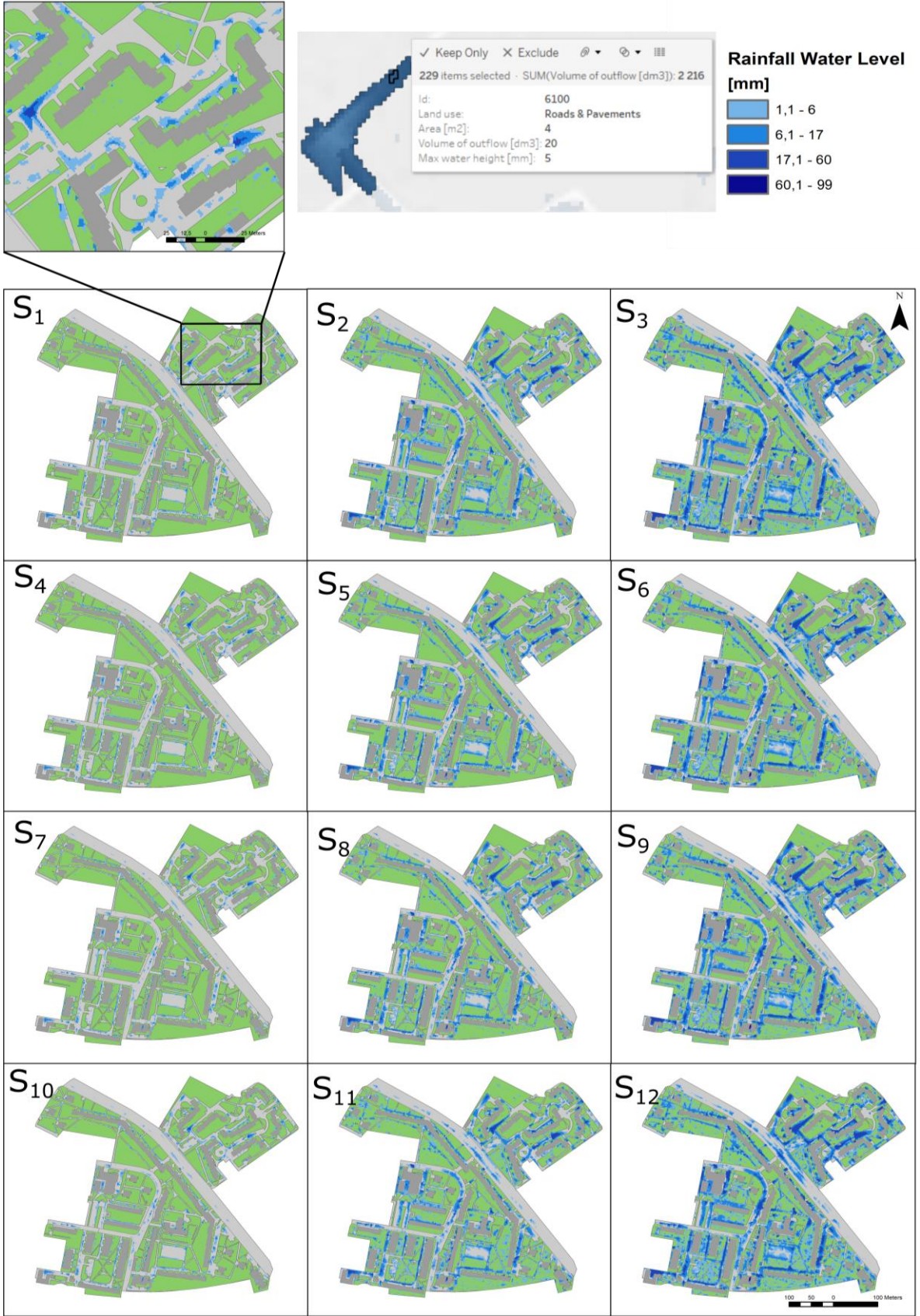

**Figure 5.** Results for stormwater accumulation scenarios.

Scenarios analysing higher precipitation values presented the same pattern as the scenarios previously commented upon. Most of the stormwater accumulation spots are identified as linear shapes along impervious communication infrastructure (roads and pavements). This makes the implementation of green infrastructure easier, as there is no need for additional surface remodelling in order to guarantee the water flow direction into the green infrastructure location (along streets and pavements). In some cases, the geometries of those spots are more rounded; however, green infrastructure implementation and land surface remodelling caused by that action will change the flow direction and the bottom point of the hydrological sink. In the case of unlikely events with the precipitation values assumed in the test of the designed system, the implementation of green infrastructure can effectively reduce the problem of stormwater accumulation.

As presented in Figure 1, the classification of green infrastructure solutions was also divided by retention ability: low (0–200 $dm^3/m^2$), medium (201–1000 $dm^3/m^2$), or high (>1000 $dm^3/m^2$). Therefore, some green infrastructure facilities are directly predesigned for specific stormwater amounts. Thus, once the specific location for the green infrastructure facility is selected, the DSS allows us to verify which type of facility is the most suitable for the water accumulation parameters. As presented in Table 3, the DSS allows us to calculate the area that is covered by water, the volume of the accumulated stormwater, and the maximum water height. These parameters can be calculated for the whole area as well as for user-defined areas, as presented in Figure 5. A comparison of the hydrological parameters measured in the DSS with the specifications of green infrastructure facilities (see Figures 1 and 2) presenting their retention abilities and sizes allows the user to select a proper solution. In a case where a few kinds of green infrastructure facilities are suitable, there are other drivers that might be taken into account by the decision-makers, like replacement costs, nuisance of use, or water purification level.

## 4.2. Additional Land Development Guidelines

While analysing the results of the model, some additional issues important in urban design were noticed. In some cases, the water flow was blocked by higher objects, which influenced the accumulation of stormwater next to those objects. In the case of the analysed study area, such objects were buildings. As a result, the accumulation of stormwater occurs next to dwellings. Such a situation may have an impact on the risk of damage to real estate. On the one hand, accumulated water may impact the stability of buildings by softening the soil profile. On the other hand, in such conditions, leakage may damage building materials and furniture, and fungus may appear due to humid conditions. In the case of windows or doors leading to basements, water can directly get inside the basement of the building, further increasing the risk of real estate damage. An example of stormwater accumulation next to a building is presented in Figure 6.

In order to reduce the risk of damage to real estate, the DSS allows for the formulation of additional land development guidelines. The results allow for the verification of stormwater accumulation from the point of view of unfavourable effects and for support with knowledge about how to remodel the land surface to avoid issues with the built environment. As the land use structure of the study area contains green areas, there are possibilities to change the water flow direction to increase the resilience of the neighbourhood. The assessment of surface regulation and its impact on safety is within the scope of statistical multicriteria analyses [88], and the proposed system completes its calculations by visualising results and considering the spatial distribution.

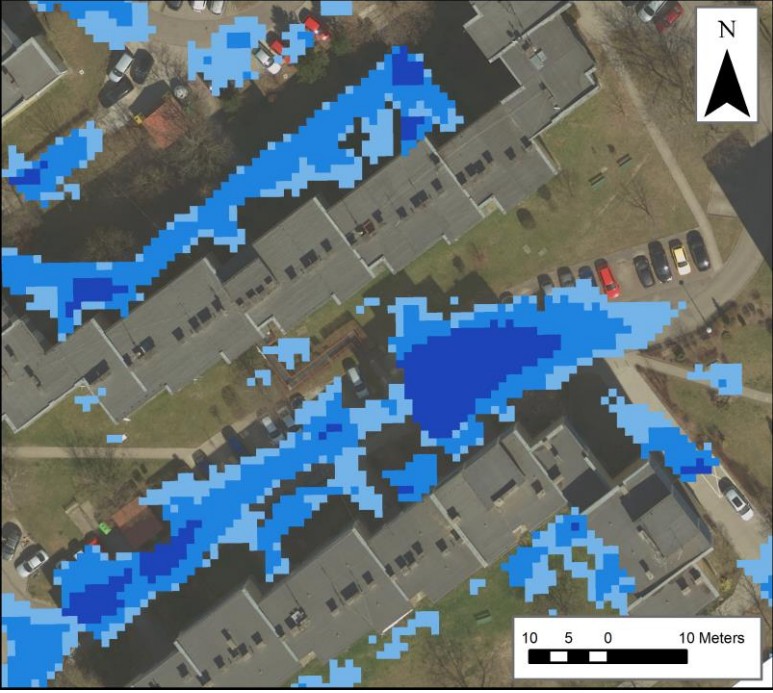

**Figure 6.** Stormwater accumulation issues.

## 5. Discussion

While searching for research in the field of stormwater monitoring, two main domains can be distinguished: stormwater pollution and stormwater volume assessment. Most papers in this field refer to the first issue; however, some research has indirectly evaluated the amount of gathered stormwater. For instance, Paule-Mercado et al. [89], by scenario stormwater runoff analysis, assessed variability in pollutant concentration, which is related with water flow and concentration itself. However, their analysis was more aggregated as one category, "urban", included buildings, spaces between those buildings, and the whole transport infrastructure. Therefore, their model is more suitable for bigger units, which makes it impossible to support green infrastructure location in the scale of one specific object. However, it is very important to highlight the use of scenario-based approaches. As BenDor et al. [60] proved in their study, the evaluation of alternative infrastructure features allows for tradeoff analysis, even leading in some cases to win–win scenarios. The usefulness of precipitation scenarios in urban management was also highlighted by Kim et al. [90]. In their investigation, scenario-based assessment was helpful in disaster mitigation. A comparison of different scenarios allowed them to quantify the relation between increasing urban green ratio and its impact on economic loss reduction. From the point of view of the mentioned studies, the scenario-based approach in the proposed system, evaluating what-if scenarios, seems to be the proper solution.

The workflow presented in this paper integrate three basic elements that have to be analysed in urban green infrastructure development policies as well as in community greening initiatives. As Xing et al. [91] summarised, these elements are (1) the identification of green infrastructure benefits (both those needed in each location and those which can be provided by each green infrastructure solution), (2) green infrastructure solutions (e.g. indoor plants, green roofs, green walls, green and blue landscaping), and (3) design decisions (including maintenance, irrigation and water management, etc.). All of those elements are combined in this research. There are similar concepts of evaluating the spatial priority of urban green infrastructure, which confirms the importance and validity of the proposed concept. Wei et al. [92] designed a solution for selecting cores with the highest connectivity which were classified as the most crucial from the point of view of biodiversity conservation and recreational use. However, their location selection system was applicable at the wider scale. The study area in their case covered 163.34 km$^2$. Therefore, the concept of selection of the most suitable location for green

infrastructure solutions applied in the designed DSS presented in this paper follows the needs also identified by other scientists.

Focusing on more detailed scale analysis, Ivanovsky et al. [93] highlighted the need for management process implementation in order to purify rainwater during runoff on soils and other covers. Again, this purification system is based on natural retention and infiltration which solves not only the issue of pollution but also the issue of water itself. As Bonneau et al. [94] investigated, in warmer periods, a significant amount of infiltrated stormwater was consumed by vegetation on slopes, even up to 100%. However, referring to exact values seems unsuitable when comparing different areas, as they depend on many local drivers that cannot be generalised. Decision-makers should not be informed by decision support systems that a specific amount of stormwater will be stored. The results should be treated with caution. In Canadian conditions, Eckart et al. [95] focused on one green infrastructure solution, such as low-impact development, employed to manage urban stormwater and restore the predevelopment hydrological conditions. Similar solutions reduced noticeably different amounts of water even when conditions seemed to be comparable. In the case of one neighbourhood, runoff volume reduction was at the level of 13%, while in a second neighbourhood, this value reached 29%. Therefore, the results obtained in Section 3 of this paper do not present specific characteristics of Polish conditions and cannot be extrapolated. In order to analyse local conditions, the proposed system analysing all variables mentioned in the method should be taken into account based on local drivers.

## 6. Conclusions

This research contributes to the state of art by way of the development of a novel DSS formed by integrating the SCS-CN method with GIS software. However, in order to present the practical aspects of the concept, the developed DSS was verified on a sample. That allowed us to answer the question of whether the application of the software is feasible with the use of real data. Therefore, our main contribution to the field is the methodological framework together with the software of the DSS itself. However, in order to increase the potential benefits of the use of the DSS, the results for the verification sample were also concluded to give an idea to urban designers about the applicability and usefulness of the software. The novelty of the presented solution lies in the combination of a location intelligence approach with a what-if approach and in the scale of the applicability of the algorithm. The DSS is universal and allows for application to any sample based on the local input parameters. The presented simulation results do not constitute a significant output themselves, but they allow for further steps in improving the situation in an analysed sample. Therefore, the research presents the process of deriving meaningful findings from geospatial information and its relationships with other objects to solve a particular problem.

The presented research shows that the integration of information about soil type, land use at the cadastral level, tree coverage, the initial soil moisture level, surface elevation, and projected precipitation levels in a DSS allows for an assessment of the accumulation of stormwater gathered in specific locations. It is important to highlight that the scale of application of the proposed DSS enables us to evaluate neighbourhoods, distinguishing inter-building space. In order to apply the system at a supra-local or regional scale, the input data type should be reconsidered. However, as heavy storms with high levels of precipitation are rather local events, the presented scale of stormwater accumulation analysis seems to be suitable. The proposed system is compatible with the commonly used SCS-CN method.

The scenario-based approach enables us to verify which locations might be more commonly flooded by stormwater and which locations are going to be flooded in unlikely cases. This provides decision-makers with the necessary information to conduct more rational planning by locating green infrastructure solutions in places where they are more greatly needed. Besides selecting proper locations, the system provides information about approximate values of accumulated water that can be managed locally. Together with examples of green infrastructure best practice, which are characterised by water retention skills, this allows for selection of the proper type of solution for each location.

Additionally, the application of the system showed that it is possible to detect property risks connected with improper land surface shape. This allows for the identification of where land surface remodelling is needed in order to reduce the risk of damage to real estate. Hence, the designed system supports environmental actions by helping in decision-making about where and how to absorb water in the built environment and, at the same time, is useful from the socio-economic point of view as it suggests cost-effective actions and warns about potential risks. Therefore, the proposed DSS fills the gap in the existing literature by presenting the concept of green infrastructure for stormwater locations. A very important fact is that the concept can be automated, which will allow the DSS to become more popular and applicable in practice. The developed DSS constitutes a technological solution based on verified and well-known methods used in hydrological modelling. However, in order to verify the accuracy of the DSS in the test area, the DSS will be verified in the future by comparing real stormwater accumulation with the results of a scenario with specific precipitation values (equal to those obtained from meteorological authorities).

The use of the designed DSS for green infrastructure location converts the decision-making process from intuitive actions to knowledge-based urban development. The DSS supports the implementation of suitable green infrastructure solutions. As a result it enables the use of a very important natural resource which is needed locally after heavy meteorological events appear. It will help in more effective urban adaptation to climate change by reducing the problem of excess water during storms and deficiency of water in dry periods. However, an open issue for local authorities is to integrate information about the amount of water together with information about its quality (especially from roads) in order to avoid damage to other natural resources (soils) when solving the problems of other resources. However, recent studies confirm that the use of constructed wetland systems with horizontal flow is effective in pollution removal [96,97] when remembering to use the proper solution adapted to local conditions [98]. Such nature- and semi-nature-based solutions can solve crucial environmental management issues in cities, leading to an urban circular economy and a closed circuit of resources [99].

**Author Contributions:** Conceptualisation, J.K.K.; methodology, J.K.K., J.C. and S.S.; software, J.C.; validation, J.K.K. and S.S.; formal analysis, J.K.K.; investigation, J.K.K.; resources, S.S.; data curation, S.S.; writing—original draft preparation, J.K.K.; writing—review and editing, J.K.K.; visualisation, J.K.K. and S.S.; supervision, J.K.K.

**Funding:** This research was funded by the Department of Spatial Economy of the Wrocław University of Environmental and Life Sciences from statutory funds.

**Acknowledgments:** The publication has been prepared as a part of the Support Programme of the Partnership between Higher Education and Science and Business Activity Sector financed by City of Wrocław.

**Conflicts of Interest:** The authors declare no conflict of interest.

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
