# Peer review of "The Development of a Novel Decision Support System for the Location of Green Infrastructure for Stormwater Management"

_sustainability, doi:10.3390/su10124388_

Round 1
Reviewer 1 Report
The authors present a manuscript on stormwater protection of urban spaces. The manuscript presents significant weaknesses that must be improved. In particular:
The title of the paper is totally misleading the reader. The paper is not dealing with Green Infrastructure in the Context of Urban Adaptation to Climate Change, but with stormwater effects reduction in urban spaces. Green Infrastructure is covering too many aspects. Is a DSS model for urban adaptation in stromwaters due to climate change. The key word is stormwater management or collection system and it is not mentioning. Thus the title should change accordingly.
The authors made a revision of the literature but they do not present the existence of adequate number of DSS systems in stormwater infrastructure implementation. What are the benefits of their proposal as all these systems are based on GIS, satellite data, historical data and other parameters that increase the DSS. This is important to verify the novelty of their proposal.
The use of the stormwater management systems presented by the authors are well established methods for many years in EU. So which is the novelty on applying these?
The models presented in this manuscript what advantages present against the ones used up to now?
The authors should add a table which will presents the numerical results and data feed to the proposed system in order to see the results. Also a comparison with the findings of other is required in order to prove the usability and the benefits of the proposed method.
Author Response
Dear Editor,
First of all I would like to thank the reviewers for their time and effort put in the review of the article. In the following paragraphs we describe how we dealt with the feedback provided.
Reviewers' comments:
Reviewer 1
The authors present a manuscript on stormwater protection of urban spaces. The manuscript presents significant weaknesses that must be improved. In particular:
The title of the paper is totally misleading the reader. The paper is not dealing with Green Infrastructure in the Context of Urban Adaptation to Climate Change, but with stormwater effects reduction in urban spaces. Green Infrastructure is covering too many aspects. Is a DSS model for urban adaptation in stromwaters due to climate change. The key word is stormwater management or collection system and it is not mentioning. Thus the title should change accordingly.
Response:
Thank you for your kind opinion according to our research.
Based on your comment the title is now changed into “The Development of a Novel Decision Support System for the Location of Green Infrastructure for Stormwater Management”. Moreover, the “stormwater management” is added to the keywords.
Reviewer 1
The authors made a revision of the literature but they do not present the existence of adequate number of DSS systems in stormwater infrastructure implementation. What are the benefits of their proposal as all these systems are based on GIS, satellite data, historical data and other parameters that increase the DSS. This is important to verify the novelty of their proposal.
The use of the stormwater management systems presented by the authors are well established methods for many years in EU. So which is the novelty on applying these?
Response:
The literature review is now enriched by more insights from articles published recently in the field of urban green infrastructure. In total there are 15 more references in the revised version. This section contains also in the current version the limitations of existing DSS in the field of stormwater management: “As Koutsoyiannis et al. [67] concluded, in their work on a decision support system for the management of water resource system, “a well-known problem in development and application of a DSS is the general lack of communication between analysts/developers and decision makers (or their technical support staff)” (lines 131-134).
The novelty of the proposed solution is now concluded as follows: “The research contributes to the state of art by the development of the novel DSS by the integration of SCS-CN method with GIS software. However, in order to present practical aspect of the concept, the developed DSS was verified on the sample. That allowed to answer the question if the application of the software is feasible with the use of real data. Therefore, the main contribution to the science is the methodological framework together with the software of the DSS itself. However, in order to increase potential benefits of the use of the DSS, the results for the verification sample are also concluded to give an idea to urban designers about applicability and usefulness of the software. The novelty of the presented solution lies on the combination of location intelligence approach with what-if approach and the scale of the applicability of the algorithm. The DSS is universal and allows for application on any sample based on the local input parameters. The presented simulation results do not constitute the significant output itself, but it allow for further steps in improving situation in analysed sample. Therefore, the research presents the process of deriving meaningful findings from geospatial information and its relationships with other objects to solve a particular problem.” (lines 443-455).
Reviewer 1
The models presented in this manuscript what advantages present against the ones used up to now?
Response:
The response for this comment is connected with previous comments. Based on the gap defined by Koutsoyiannis et al. there is a lack of linkages between application developers/specialists in environmental modelling and decision makers/practitioners. Therefore, “most of the DSS for water management operate on a scale that allows to describe water management processes in a wider context. Therefore they enable to identify hot-spots of stormwater issues in a district of the city [67,69]. However, they fail to analyse local conditions, describing the situation in details with the high precision of input data obtained results. Therefore, practise is still facing the problem of lack of the local solutions which would refer not to the holistic approaches for bigger area, but everyday decisions that have to be made by public services units.” (lines 137-143).
Our solution is “the combination of location intelligence approach with what-if approach and the scale of the applicability of the algorithm. The DSS is universal and allows for application on any sample based on the local input parameters. The presented simulation results do not constitute the significant output itself, but it allow for further steps in improving situation in analysed sample. Therefore, the research presents the process of deriving meaningful findings from geospatial information and its relationships with other objects to solve a particular problem.” (lines 450-455).
Reviewer 1
The authors should add a table which will presents the numerical results and data feed to the proposed system in order to see the results. Also a comparison with the findings of other is required in order to prove the usability and the benefits of the proposed method.
Response:
Thank you for that remark. The results are now also presented in Table 3, which indeed is valuable and allows for better understanding the green infrastructure facility selection based on the simulation results.
The developed DSS is discussed in relation to few other systems in the Discussion (section 5). However, this section is now expanded and additional studies are analysed:
- “The usefulness of precipitation scenarios in urban management was highlighted also by Kim et al. [90]. As they investigated, the scenario-based assessment was helpful in disaster mitigation. Comparison of different scenarios allowed to quantify the relation between increasing urban green ration and its impact on economic losses reduction.” (lines 405-408).
- “The workflow presented in this paper integrate three basic elements that have to be analysed in urban green infrastructure development policies as well community greening initiatives. As Xing et al. [91]summarised, these elements are: (1) the identification of green infrastructure benefits (both needed in each location as well as those benefits which can be provided by each green infrastructure solution), (2) green infrastructure solution (e.g. indoor plants, green roofs, green walls, green & blue landscaping), and (3) design decisions (including maintenance, irrigation and water management, etc.). All those elements are combined in this research. There are similar concepts of evaluating spatial priority of urban green infrastructure, which confirms the importance and validity of the proposed concept. Wei et al. [92] designed the solution for selecting cores with the highest connectivity which were classified as the most crucial from the point of view of biodiversity conservation and recreational use. However, their location selection system was applicable at the wider scale. The study area covered in their case 163.34 km². Therefore, the most suitable location selection concept for green infrastructure solution, applied in the designed DSS presented in this paper, follows the needs identified also by other scientists.” (lines 411-424).
Reviewer 2 Report
The paper introduced a Decision Support System to identify the location of urban hydrological sinks and the amount of water. The DDS is implemented in GIS and can be applied for the planning of green infrastructure. The topic is interesting, and the proposed DDS can benefit sustainable urbanization.
However, the details of the implementation of the DDS is oversimplified and need more elaboration.
More specific comments:
1. The introduction section is too long and should be more concise. The mentioned examples can be moved to a separate section, e.g., a literature review section.
2. The paper fails to mention how to evaluate the accuracy of the proposed DSS. Is empirical data available to evaluate the accuracy of the calculated water height under each scenario?
3. How to integrate different data format in the DDS. The DDS utilizes raster data, vector data, and LiDAR data. How the different formats data can be integrated into the calculation of water height?
4. Most importantly, what specific calculations are performed at each step in Fig3? The DDS is currently a black box. E.g., what are ACM II and ACM I? Equation 3 on line 224 doesn't make sense. What does 25,4 mean? The calculation of precipitation runoff needs more explanation.
5. Where are the results of S1-3 in Fig 5?
Author Response
Dear Editor,
First of all I would like to thank the reviewers for their time and effort put in the review of the article. In the following paragraphs we describe how we dealt with the feedback provided.
Reviewers' comments:
Reviewer 2
The paper introduced a Decision Support System to identify the location of urban hydrological sinks and the amount of water. The DDS is implemented in GIS and can be applied for the planning of green infrastructure. The topic is interesting, and the proposed DDS can benefit sustainable urbanization.
However, the details of the implementation of the DDS is oversimplified and need more elaboration.
1. The introduction section is too long and should be more concise. The mentioned examples can be moved to a separate section, e.g., a literature review section.
Response:
Thank you for your kind opinion according to our research.
Based on your comment the literature review section is now separated.
Reviewer 2
2. The paper fails to mention how to evaluate the accuracy of the proposed DSS. Is empirical data available to evaluate the accuracy of the calculated water height under each scenario?
Response:
New subsection is now added describing input data, their sources as well as accuracy: “In order to support the designed DSS to calculate each case, there is a need to supply following data:
1. Soil types – in case of verification sample: Geological and engineering atlas of the Wrocław agglomeration [72]. The geological and engineering atlas presenting soil type was prepared based on thematic maps in scale 1:10 000.
2. Land use – in case of verification sample: real estate cadastre register kept by the Municipal Board of Geodesy, Cartography and Cadastre. The accuracy of coordinates in that database is 0,10 m.
3. Tree coverage and elevation – in case of verification sample: airborne laser scanning LiDAR for urban areas. In the case of LiDAR, it is possible to achieve a 0.5 m x 0.5 m resolution, however, in case of this research the 1.0 m x 1.0 m grid was constructed.” (lines 180-188).
Moreover, in Conclusions we highlighted that: “The developed DSS constitute a technological solution based on verified and well-known methods used in the hydrological modelling. However, in order to verify the accuracy of the DSS in test area, the DSS will be verified in the future by comparing real stormwater accumulation with the results of a scenario with specific precipitation value (equal to those obtained from meteorological authorities).” (lines 447-451).
Reviewer 2
3. How to integrate different data format in the DDS. The DDS utilizes raster data, vector data, and LiDAR data. How the different formats data can be integrated into the calculation of water height?
Response:
In section 3.1. the information about data conversion and pre-processing are now included: “All data were used in the developed Python script for ArcGIS. All input data used in the program has been converted to a vector format which is the basic data format used by the program. To integrate the values required in the model from different input vector data served their mutual spatial relations.
At the stage of pre-processing of input data from the LiDAR point cloud data, measurement points corresponding to the soil class and buildings were selected. Next, based on the average value of points in a grid with dimensions of 1x1 m was made to the raster with the same pixel size. The resulting Digital Elevation Model (DEM) was then used as input to the iterative process of surface runoff analysis.” (lines 189-197).
Reviewer 2
4. Most importantly, what specific calculations are performed at each step in Fig3? The DDS is currently a black box. E.g., what are ACM II and ACM I? Equation 3 on line 224 doesn't make sense. What does 25,4 mean? The calculation of precipitation runoff needs more explanation.
Response:
All calculations and analyses are explained in details in the following subsections (line 164). Each subsection has the same name as the block in the diagram on Figure 3. We also added the information that whole SCS-CN method is explained in details in previous papers (line 165).
Moreover, additional details are included in the revised version of the paper:
· about AMC: “The AMC is a variable which is calculated in the DSS. Based on the field studies which gave the basis for the SCS-CN method, three different AMC are defined in the standardized tables. Therefore, AMC influenced the selection of variables which were taken from the standardized table [74].” (lines 263-266);
· about 25,4: ” 25,4 – conversion rate from English units to International System of units (from inches to millimetres)” (lines 282-283); moreover, the equation comes directly from the SCS-CN method;
· about precipitation runoff is now more elaborated: “The precipitation runoff was carried out in automatic iteration manner. As mentioned in section 3.1, at the stage of pre-processing of input data from the LiDAR point cloud data, measurement points corresponding to the soil class and buildings were selected. Only such classes were able to draw out and constituted a physical barrier water surface runoff. Then based on the average value of points in a grid with dimensions of 1x1 m was made to the raster with the same pixel size. This DEM was used as input to the iterative process of surface runoff analysis. The effective precipitation was flowing on DEM surface depending on slope direction and was filling out each hydrological sink. The DEM input grid performed a multiple task, firstly the vector layer generated on it from the centroid of individual pixels is information about the altitude value for the first interception. Once a hydrological sink was full (the water level reached edges of each sink, the system created a new DEMi , which excluded that hydrological sink from further iteration steps. At this stage that hydrological sink was covered with plane surface and allowed to flow stormwater further to next sinks according to DEM flow direction. Those steps were repeated until effective precipitation has been used. Finally, precipitation runoff DEMs constituted the output for the volume of stormwater calculation.” (subsection 3.10.).
Reviewer 2
5. Where are the results of S1-3 in Fig 5?
Response:
Scenarios S1-S3 were prepared as a verification to other scenarios with tree coverage. However, in order to show these differences to all readers S1-S3 are now added to the figure. Thank you for that comment as it is indeed more reasonable solution.
Reviewer 3 Report
Thank you for submitting your manuscript to the sustainability journal. The research topic fits very well with the scope of the sustainability journal. Nevertheless, there are some remarks regarding the manuscript for further improvement. First of all, from my point of view, it should not be classified as an article but a case study. The structure of the manuscript does not respect Scientific Best Practice in terms of structuring a scientific publication (introduction – site description – materials and methods – results – discussion – conclusions). Further, there are detailed remarks.
1. Introduction
There is already provided a general overview on the existing literature. Anyhow, I strongly recommend to review the publications of the Sustainability journal as there have already been several studies published in the last years regarding urban green infrastructure.
There must be given a clear scope of the manuscript. The scope is to be concluded from gaps in the exististing literature.
Further, for the figures 1 and 2 must be obtained the copyright permission.
2. Methodological framework
After the introduction section should follow the materials and methods section, where the site description might be part of it in order to provide all necessary data to the reader to follow the text. The materials and methods section is missing, which is a deficit, as the reader does not have any chance to understand a) what is the scope of the investigation and b) where is the innovation potential of the contribution.
It remains unclear, if the described approach is only the framework or if it was in that way applied to the particular case study. For that reason, the section should be renamed "materials and methods" and treated as such to provide the clear research approach. Further, the site desciprtion of the site is missing.
3. Results
It should be illustrated which options existed to be evaluated through a decision support system.
4. Discussion
The discussion is by far too short.
5. Conclusions
The DSS and it' s application in that particular case remained unclear to the reader.
Author Response
Dear Editor,
First of all I would like to thank the reviewers for their time and effort put in the review of the article. In the following paragraphs we describe how we dealt with the feedback provided.
Reviewers' comments:
Reviewer 3
Thank you for submitting your manuscript to the sustainability journal. The research topic fits very well with the scope of the sustainability journal. Nevertheless, there are some remarks regarding the manuscript for further improvement. First of all, from my point of view, it should not be classified as an article but a case study. The structure of the manuscript does not respect Scientific Best Practice in terms of structuring a scientific publication (introduction – site description – materials and methods – results – discussion – conclusions). Further, there are detailed remarks.
Response:
Thank you for your kind opinion according to the article. About the type of papers, the presented decision support system (DSS) is the universal solution that can be applied in aby other location. However, in order to verify the DSS in was verified on a sample represented by study area for which detailed characteristics were obtained. That allowed to insert specific parameters for calculation for the DSS verification. Moreover, due to the Instructions for Authors on the website of Sustainability there are only two types of articles that can be selected: Articles and Reviews. As the paper presents the system designed by the Authors, it should not be classified as Review.
According to the comment about the structure of the manuscript, the structure was modified based on suggestions of both Reviewers.
Reviewer 3
1. Introduction
There is already provided a general overview on the existing literature. Anyhow, I strongly recommend to review the publications of the Sustainability journal as there have already been several studies published in the last years regarding urban green infrastructure.
There must be given a clear scope of the manuscript. The scope is to be concluded from gaps in the exististing literature.
Further, for the figures 1 and 2 must be obtained the copyright permission.
Response:
The literature review was enriched by more insights from articles published recently in the field of urban green infrastructure. In total there are 15 more references in the revised version. Based also on the suggestion of the Reviewers 1, there is a separate section focused on literature review.
Thank you very much for your good suggestion according to clarifying gap of knowledge and scope of the research. It was not emphasized well in previous version. Based on the literature review, the gap in existing literature as well as scope of the research is now stated in lines 131-136.
The permission for copyrights of Figures 1 and 2 was obtained from the City Council of Wrocław under the condition of displaying the original source, which in mentioned in brackets in descriptions of those figures.
Reviewer 3
2. Methodological framework
After the introduction section should follow the materials and methods section, where the site description might be part of it in order to provide all necessary data to the reader to follow the text. The materials and methods section is missing, which is a deficit, as the reader does not have any chance to understand a) what is the scope of the investigation and b) where is the innovation potential of the contribution.
It remains unclear, if the described approach is only the framework or if it was in that way applied to the particular case study. For that reason, the section should be renamed "materials and methods" and treated as such to provide the clear research approach. Further, the site description of the site is missing.
Response:
New structure of the paper is applied – based on your comments on the comments of Reviewer 1. There is also new subsection on “Verification sample and data” (subsection 3.1.) which structure all external data what were used while verification of the DSS. The information about necessary data to support the system is now joined with databases used in case of the verification, as well as the accuracy on these data.
In order to make it clear that the research presents the framework of the DSS, and to show the innovation potential, the following sentence was added: “The scope of this study is to integrate the best practises from the hydrological and urban domains. The development of the novel DSS constitutes the innovation to the urban design domain by the applicable solution which unified software which integrates verified and well recognized hydrological method (SCS-CN) among specialists in water management field, with GIS-based analyses which are commonly used in the domain of urban studies.” (lines 156-161).
Reviewer 3
3. Results
It should be illustrated which options existed to be evaluated through a decision support system.
Response:
The Figure 5 was changed and now it presents in zoom the example of possible measurement of the hydrological features of accumulated stormwater.
Reviewer 3
4. Discussion
The discussion is by far too short.
Response:
The discussion is now expanded and it refers to more studies and their methods, applications and outcomes. This section includes right now also additional papers that you suggested in previous comment – articles on urban green infrastructure from Sustainability journal.
Reviewer 3
5. Conclusions
The DSS and it' s application in that particular case remained unclear to the reader.
Response:
Thank you very much for that remark. Indeed, the previous version could be unclear as one paragraph focuses on the system, while the second focuses on the results from the verification sample. Therefore, the additional paragraph is added in the new version to make in clear. “The research contributes to the state of art by the development of the novel DSS by the integration of SCS-CN method with GIS software. However, in order to present practical aspect of the concept, the developed DSS was verified on the sample. That allowed to answer the question if the application of the software is feasible with the use of real data. Therefore, the main contribution to the science is the methodological framework together with the software of the DSS itself. However, in order to increase potential benefits of the use of the DSS, the results for the verification sample are also concluded to give an idea to urban designers about applicability and usefulness of the software.” (lines 416-422).
Moreover, according to language changes, the manuscript was edited by licensed interpreter, which has a TOLES Advanced Certificate.
Round 2
Reviewer 2 Report
If 25,4 is the conversion rate, please use 25.4 instead of 25,4 to avoid confusion.
Author Response
Reviewer 1
If 25,4 is the conversion rate, please use 25.4 instead of 25,4 to avoid confusion.
Response:
Thank you for your remark. The conversion rate was changed into 25.4.
Reviewer 3 Report
Thank you for providing the revised version of the manuscript. It was improved significantly. From my point of view, still there is a need to better describe the decision finding procedure and how it was implmented in the DSS.
Author Response
Reviewer 1
Thank you for providing the revised version of the manuscript. It was improved significantly. From my point of view, still there is a need to better describe the decision finding procedure and how it was implemented in the DSS.
Response:
Thank you for your comment. In order to better present the abilities of the DSS, the Table 3 is now added. Together with improved Figure 5 (with window enabling measuring specific place) it presents the values that can be calculated by the system.
Moreover, additional description explaining how the outcomes of the DSS can be useful for green infrastructure selection is now included in the paper: “As presented in Figure 1, the classification of green infrastructure solutions was divided also due to the retention abilities: low (0-200 dm³/m2), medium (201-1000 dm³/ m2), high (>1000 dm³/m2). Therefore, some green infrastructure facilities are directly predesigned for specific stormwater amounts. Therefore, once the specific location for green infrastructure facility is selected, the DSS allows to verify which type of facility is the most suitable due to the water accumulation parameters. As presented in Table 3, the DSS allows to calculate the area that is covered by water, the volume of the accumulated stormwater and the maximum water height. These parameters can be calculated for whole area as well as for user-defined area, as presented in the Figure 5. Comparison of the hydrological parameters measured in the DSS with the specification of green infrastructure facilities (see Figure 1 and 2) presenting retention abilities and sizes, allows to select by the user proper solution. In case of the suitability of few kinds of green infrastructure facilities, there are other drivers that might be taken into account by the decision makers, like replacement costs, nuisance of use or water purification level.” (lines 359-371).